# Long- and Short-Term Conductance Control of Artificial Polymer Wire Synapses

**DOI:** 10.3390/polym13020312

**Published:** 2021-01-19

**Authors:** Naruki Hagiwara, Shoma Sekizaki, Yuji Kuwahara, Tetsuya Asai, Megumi Akai-Kasaya

**Affiliations:** 1Graduate School of Engineering, Osaka University, Osaka 565-0871, Japan; hagiwara@ss.prec.eng.osaka-u.ac.jp (N.H.); sekizaki@ss.prec.eng.osaka-u.ac.jp (S.S.); kuwahara@prec.eng.osaka-u.ac.jp (Y.K.); 2Faculty of Information Science and Technology, Hokkaido University, Sapporo 060-0814, Japan; asai@ist.hokudai.ac.jp

**Keywords:** artificial synapse, conductive polymer wire, PEDOT:PSS, resistance change memory

## Abstract

Networks in the human brain are extremely complex and sophisticated. The abstract model of the human brain has been used in software development, specifically in artificial intelligence. Despite the remarkable outcomes achieved using artificial intelligence, the approach consumes a huge amount of computational resources. A possible solution to this issue is the development of processing circuits that physically resemble an artificial brain, which can offer low-energy loss and high-speed processing. This study demonstrated the synaptic functions of conductive polymer wires linking arbitrary electrodes in solution. By controlling the conductance of the wires, synaptic functions such as long-term potentiation and short-term plasticity were achieved, which are similar to the manner in which a synapse changes the strength of its connections. This novel organic artificial synapse can be used to construct information-processing circuits by wiring from scratch and learning efficiently in response to external stimuli.

## 1. Introduction

The human brain was first described as a complex network of innumerable neurons connected via synapses in the early 20th century. This discovery inspired various mathematical models for information processing, including models by McCulloch and Pitts [1], which have contributed towards the development of artificial intelligence (AI) technology using conventional von Neumann-type computers. However, the application of AI software leads to issues regarding energy loss and execution processing time due to the structure of these conventional computers, especially as the network models become larger and more complex [2]. The execution of advanced and highly efficient information processing similar to that performed by the human brain involves complicated and elaborate three-dimensional networks, where a new processor should be developed to replace the conventional computer.

New processers based on the concept of neuromorphic hardware have emerged in recent years, where several studies have aimed to reproduce the functions of brain synapses by skillfully utilizing the unique electrical properties of various materials [3,4]. Specifically, synaptic devices have been developed using poly(3,4-ethylenedioxy-thiophene) doped with poly(styrene sulfonate) anions (PEDOT:PSS) [5,6], which is a widely-used conductive polymer that offers high conductivity [7], thermoelectric conversion [8,9], and chemical sensitivity with bio-adaptability [10,11]. The PEDOT cation and the dopant PSS anion are electrostatically bonded, where PSS pull electrons from the PEDOT chain and inject positive carriers (Figure 1a). This property leads to high conductivity. PEDOT:PSS has a hierarchical structure [12], where the state of each structure has a significant influence on the overall morphology and electrical conductivity.

PEDOT:PSS has been previously grown into the shape of a wire during electrodeposition under a bipolar square-wave alternating current (AC) voltage between two electrodes immersed in a precursor solution of 3,4-ethylenedioxythiophene (EDOT) monomer and PSS [13,14,15,16]. The resulting PEDOT:PSS wire was used to construct information-processing circuits [17]. The electropolymerization of PEDOT proceeds on the surface of the anode immersed in the precursor solution, where PSS becomes incorporated as a dopant to deposit PEDOT: PSS with a higher-ordered structure. The morphology of the resulting PEDOT:PSS can be controlled by adjusting the growth voltage conditions [18,19]. For example, either dendritic or wire-shaped PEDOT:PSS may be produced, as shown in Figure 1b,c, respectively (experimental details given in Appendix A). In particular, wire-shaped PEDOT:PSS offers excellent wiring properties due to elongation of the polymer wires along the electric field during anisotropic polymerization. This allows for the formation of an electric circuit network with the PEDOT:PSS wire between arbitrary electrodes (Figure 1d). The appearance of polymer wires growing and linking between electrodes is similar to the ~100 billion neurons that form independently in a newborn human brain, which comprises axons extending to other neurons to form a neural circuit network [20]. Therefore, polymer wires show promise for easily and inexpensively producing information-processing circuits with three-dimensional and spatial integration similar to an actual human brain.

A previous study by the current authors demonstrated an increase in conductance value between electrodes due to linking of conductive polymer wires to achieve synaptic long-term potentiation (LTP) [21,22], which was applied in a synaptic device [17]. General electronic synaptic devices reproduce the change in synaptic strength by adjusting the resistance [23,24,25]. However, PEDOT:PSS is a stable material, and an effective way to selectively depolymerize the wires after linking has not yet been reported. In other words, previous reports on our synaptic device were able to increase the conductance value between the electrodes, but not decrease it. Therefore, a network constructed by wiring is unable to learn flexibly according to the surrounding environment after network formation.

This study aimed to demonstrate the use of bridging wires to control the conductance between two electrodes, thereby emulating the way in which a real synapse changes its strength of connection with another neuron. Conductance modification was achieved by controlling the diameter of the bridging wires and reproducing a synaptic LTP function. Synaptic short-term plasticity (STP) describes a temporary change in synaptic connection strength [26,27], which was also achieved by controlling the conductivity of the bridging wires. This STP characteristic allowed for a temporary reduction in the conductance between the electrodes, which has not yet been reported. The resulting artificial polymer wire synapses show promise for application in highly-integrated information-processing circuits, as they can be formed from scratch, select input information to memorize, and efficiently implement learning.

## 2. Materials and Methods

The growth and linking of the polymer wires were conducted between the tips of an electrode gap, where the conductance was adjusted based on continuous conductance readings (Figure 2). The applied voltage control and conductance measurements were based on mechanical relay control using a microcomputer Arduino DUE (Arduino S.R.L., Torino, Piemonte, Italy). An electrode gap of 50 μm was fabricated by patterning a glass substrate via photolithography and the deposition of Cr (10 nm) and Au (190 nm) using EB1100 (Canon Anelva, Kawasaki, Kanagawa, Japan). The precursor solution contained 0.135 M EDOT and 0.02 M PSS (Sigma-Aldrich, St. Louis, Missouri, USA) in a mixture of acetonitrile and ultrapure water (1:1). A polydimethylsiloxane (PDMS) cavity was fixed around the tips of the electrode gap and filled with the precursor solution. A growth voltage was applied to the Au electrodes immersed in the precursor solution to facilitate electrodeposition of the conductive polymer wires at the electrode and wire tips, where the electrodes became linked with the polymer wires. The growth voltage was a bipolar square-wave AC voltage generated using an arbitrary waveform generator WF1973 (NF Corporation, Yokohama, Kanagawa, Japan) connected to a bipolar amplifier HSA4101 (NF Corporation, Yokohama, Kanagawa, Japan). The read direct current (DC) voltage of −0.1 V and voltage pulses for the conductance modification were generated from the analog output pin of the microcomputer, which was connected to a self-made voltage amplifier capable of offset control. In-situ optical microscopy was conducted from beneath via a glass substrate using an inverted optical microscope IX73 (Olympus, Shinjuku, Tokyo, Japan) (Figure 1b–d). Optical microscope images of the dried samples were acquired from the top using a digital microscope VHX-500 (KEYENCE, Osaka, Osaka, Japan). Atomic force microscopy (AFM) images of the dried samples were obtained by SPA 400 (SII Nano Technology, Chuo, Tokyo, Japan) in dynamic force mode.

Polymer wire growth was conducted under a bipolar square-wave AC voltage (50 kHz, 20 V_p-p_) with a 50% duty cycle applied to the working electrode, where the counter electrode was switched to ground (GND). The conductance value between the electrodes was measured at 4-s intervals, which consisted of a 3.5-s wire growth phase and a 0.5-s conductance reading phase. During the reading phase, the counter electrode was switched to a current–voltage (I–V) amplifier with a magnification of 2 × 10^6^. The number of bridging wires was determined using in-situ optical microscopy. The polymer wire growth was manually stopped based on the number of bridging wires and their conductance.

Conductance modification was achieved by applying varying voltage pulses with an amplitude of *V*, width of *W*, and repetition interval of *T* to the working electrode, where the counter electrode was switched to GND. During the interval of pulses, the applied voltage was −0.1 V and the counter electrode was switched to the I–V amplifier to measure the conductance value between the two electrodes. The amplified voltage signal was sent from the I–V amplifier to an analog input pin of the microcomputer, and the conductance value was recorded and displayed on the PC monitor.

## 3. Results

### 3.1. Expression of LTP

An optical microscope image of the bridging polymer wires was acquired when the conductance value between electrodes was 1.45 µS (Figure 3a). Then, we applied periodic pulses with *V*, *W*, and *T* values of 2.5 V, 10 ms, and 5 s, respectively to the electrode gap. Figure 3b shows the change in conductance during 30 pulses were applied from 20 to 170 s. Immediately after a pulse was applied, a peak conductance value appeared due to the ion current, after which it converged to a net conductance value between the electrodes due to the ion current stopping for a few seconds. Specifically, the initial conductance value of 1.45 µS dropped to 0.15 µS after the first voltage pulse was applied. Dedoping of PSS from the polymer significantly decreased the conductance of the entire polymer wire near the cathode, while electropolymerization simultaneously proceeded on the surface of the polymer wire near the anode. Thus, the conductance value increased as the pulses were continuously applied. Dedoping did not proceed during further pulse application, while polymerization proceeded at every pulse to gradually increase the conductance of the entire wire. Consequently, the conductance after the 30th pulse was 3.30 µS higher than that immediately after the first pulse. This increased conductance was retained semi-permanently, where no conductance change was observed after 1 d in solution and 6 months in dried samples stored in ambient atmosphere. This change in conductance indicated that the LTP synaptic function had been reproduced, where the strength of a synaptic connection was enhanced for an extended period to form long-term memory by repeated stimulation. A control experiment was conducted using an electrode gap without any bridging polymer wires, where the net conductance between the electrodes remained at ≈0 µS during continuous pulsing, and LTP was not observed (inset in Figure 3b). This confirmed that LTP was attributed to the conductance change of the bridging polymer wires.

The change in conductance was also evaluated using continuous voltage pulses with intervals of *T* = 2 and 60 s (Figure 3c,d). LTP was observed clearly at an interval of 2 s, where the conductance increased by 7.25 µS compared over 30 pulses. However, the net conductance value between the electrodes remained low during continuous voltage pulsing with a 60 s interval, and LTP was not observed. Optical microscopy revealed that pulses with long intervals led to uniformly thicker wires (Appendix A), but no conductance increase was observed. On the other hand, the polymer wires that exhibited LTP had an asymmetric diameter distribution in the longitudinal direction after pulsing. The AFM height profiles and optical microscope images of the polymer wires linking the electrode gap before and after 100 periodic voltage pulses (*V* = 2.5 V, *W* = 10 ms, *T* = 2 s) revealed a significant increase in wire diameter near the anode during voltage pulsing, compared to that near the cathode (Figure 3e,f). These results indicated that the sufficiently high conductive PEDOT:PSS was polymerized on the surface of the polymer wires near the anode by applying voltage pulses, which increased the conductance of the entire wire and imparted sufficient robustness of the wire shape for stable long-term conductance.

### 3.2. Expression of STP

The application of bipolar voltage pulses with a lower amplitude to the asymmetric polymer wires acquired by the expression of LTP induced a further change in conductance, which included both an increase and decrease (Figure 4a). The change in conductance between the electrodes during continuous voltage pulsing (*V* = 0.9 or −1.5 V, *W* = 10 ms, and *T* = 0.5 s) was observed as the polarity was changed between negative and positive every 200 pulses. The conductance gradually increased with the continuous application of positive pulses, and gradually decreased when the polarity of the applied pulses was negative. This change in conductance continued after repeated switching, demonstrating that the artificial polymer wire synapse had a high switching repeatability. Thus, the conductance can be consecutively and reversibly modified by controlling the number and polarity of the applied voltage pulses. As the polymer wire synapse was capable of reproducing both potentiation and depression, it may be regarded as an artificial synapse with plasticity due to its ability to repeatedly rewrite its conductance value. This conductance control has never been successfully achieved using normal polymer wires with a uniform thickness, and was found to be dependent on an asymmetric polymer wire shape.

The retention characteristics of the polymer wire synapse in the low-resistance state (LRS) and high-resistance state (HRS) were evaluated, where the change in conductance over time with an open-circuit was observed (Figure 4b). The initial conductance values in LRS and HRS decayed over a few seconds until the LRS and HRS conductance values converged to the same value. Thus, the polymer wire synapse could increase or decrease its conductance for a short time by applying small voltage pulses, and could reproduce synaptic STP functions that form short-term memory.

This STP characteristic was observed in both the precursor solution containing EDOT and PSS and the dopant PSS aqueous solution without the EDOT monomer. We conducted control experiments with different concentrations of the PSS solutions (0.1, 0.01, 0.001 M, or ultrapure water). First, the polymer wires were linked between the tips of the electrode gap via electrodeposition in the precursor solution, and the shape of the wires became asymmetric as a periodic voltage pulse (*V* = 2.5 V, *W* = 10 ms, *T* = 2 s) was applied 100 times. Then, the sample was removed from the precursor solution, dried, and immersed in a PSS aqueous solution. While attempting to achieve the STP characteristic by applying small voltage pulses (*V* = 0.9 or −1.5 V, *W* = 10 ms, *T* = 0.5 s), the conductance change was found to be less likely to occur at a lower PSS concentration (Figure 4c). Thus, the bidirectional change in conductance was attributed to reversible doping and dedoping derived from the PSS anion.

## 4. Discussion

LTP expression was affected by the interval of voltage pulse application, as the interval had an effect on the diameter distribution of the polymer wires and the conductivity of the newly-polymerized polymer film on the wire surface. These differences were attributed to the difference in the PSS anion distribution of the solution according to the voltage pulse frequency. The frequent application of voltage pulses led to increased attraction of the PSS anions towards the anode along the line of electric force, which led to a high density of PSS anions near the anode. This facilitated the electrodeposition of PEDOT:PSS with a higher-order structure, where highly conductive PEDOT:PSS containing abundant dopant PSS molecules was deposited on the wire surface near the anode. Further, the insufficient PSS anion density further from the anode led to minimal electrodeposition. This localized increase in wire diameter during electrodeposition led to an asymmetric wire shape and increased conductance, similar to LTP. Less-frequent voltage pulsing allowed for longer diffusion of the migrated PSS anions between the applied pulses, leading to more uniform and less-dense distribution throughout the wire. Consequently, a low-conductivity polymer containing a small amount of dopant was deposited on the entire wire surface, resulting in a low conductance between the electrodes.

In the STP expression experiment, the asymmetric wire shape and reversible doping reaction were the driving principles of the doping/dedoping reaction given in Equation (1):(1)$${PEDOT}^{+}:{PSS}^{-}\;+\;e^{-}\;⇋{PEDOT}^{0}+{PSS}^{-}$$

The application of a voltage to the electrode gap linked by polymer wires led to simultaneous doping near the anode and dedoping near the cathode. These reactions would cancel one another in a typical two-terminal system, thereby minimizing the variation in conductance along the entire wire before and after applying voltage. However, an asymmetric diameter distribution along the longitudinal direction of the wire led to disproportionate doping efficiency at the anode side and dedoping efficiency at the cathode side due to the difference in the wire surface area around the anode and cathode. Consequently, the total number of carriers varied along the wire, leading to a change in conductance before and after applying voltage.

The changes in conductance caused by the doping and dedoping reactions decayed and converged to the same value within a period of a few tens of seconds to a few minutes. This may have been attributed to the uniform concentration gradient achieved due to PSS anion diffusion over time. The application of voltage to the electrode gap immersed in the PSS solution generated electrostatic forces that caused PSS anions to aggregate near the anode, which enhanced the doping reaction. When the voltage was halted, the PSS anions that had not engaged in the doping reaction diffused readily, while the PSS anions engaged in doping dissociated more gradually. Consequently, the concentration gradient in the electrode gap became uniform over time. Considering the effect of Le Chatelier’s principle on Equation (1), the change in the concentration gradient of PSS anions led to a new equilibrium state of the wire near the electrodes. The reverse reaction was promoted due to diffusion, thereby returning the conductance to the initial value.

The polymer wire synapse allowed for easy LTP expression changes according to the frequency of the applied voltage pulses, thereby reproducing the multistore model of human memory [28]. This property allows for the selection of frequently encountered input information for long-term memory, thereby facilitating efficient learning. In addition, the polymer wire synapse expressed STP when small voltage pulses were applied after LTP expression (Appendix A). This allowed for rewriting after the formation of a long-term memory, demonstrating the potential for flexible learning according to the surrounding environment. Synaptic devices with STP characteristics have gained recent popularity, and can be applied to information-processing models that reflect past input information for a certain period of time, such as recurrent neural networks (RNNs) [29]. For example, the implementation of reservoir computing (RC) [30], which is a type of RNN model, has been previously demonstrated in hardware using memristors with resistance-changing properties, such as STP [31].

## 5. Conclusions

This study facilitated conductance changes between two electrodes linked by conductive polymer PEDOT:PSS wires via the continuous application of voltage pulses to the electrode gap, thereby achieving synaptic functions including LTP and STP. The learning efficiency and retention time were dependent on the frequency and magnitude of the input voltage pulses, thus the system may be regarded as an artificial synapse capable of learning efficiently via selective storage of input information. The polymer wire synapse can be used to construct an information-processing network circuit from scratch and learn efficiently in response to external stimuli. This shows promise for future applications in highly integrated brain-type hardware to reproduce the structure and learning mechanism of the human brain.

## Figures and Tables

**Figure 1 polymers-13-00312-f001:**
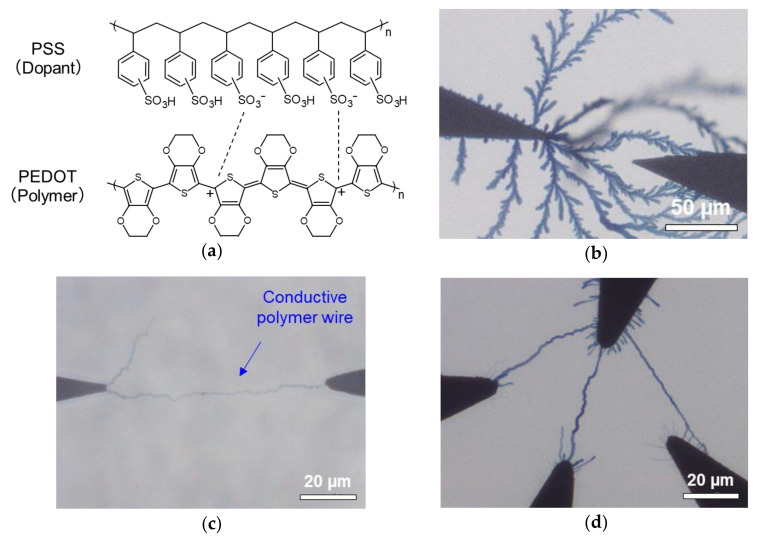
(**a**) Molecular structure of poly(3,4-ethylenedioxy-thiophene) doped with poly(styrene sulfonate) anions (PEDOT:PSS) conductive polymer. (**b**) Dendritic and (**c**) wire- shaped PEDOT:PSS obtained via electrodeposition. (**d**) Network circuit formed via branch wiring of PEDOT:PSS polymer between four electrodes.

**Figure 2 polymers-13-00312-f002:**
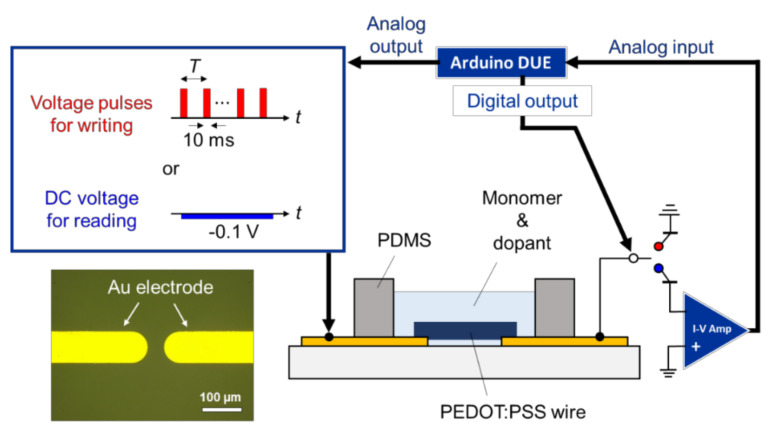
Schematic diagram of the experimental setup with a corresponding optical microscope image of the Au electrode gap. During conductance writing, the microcomputer supplied continuous voltage pulses and the counter electrode was switched to ground (GND). During conductance reading, the microcomputer supplied a −0.1 V DC voltage and the counter electrode was switched to the I–V amplifier.

**Figure 3 polymers-13-00312-f003:**
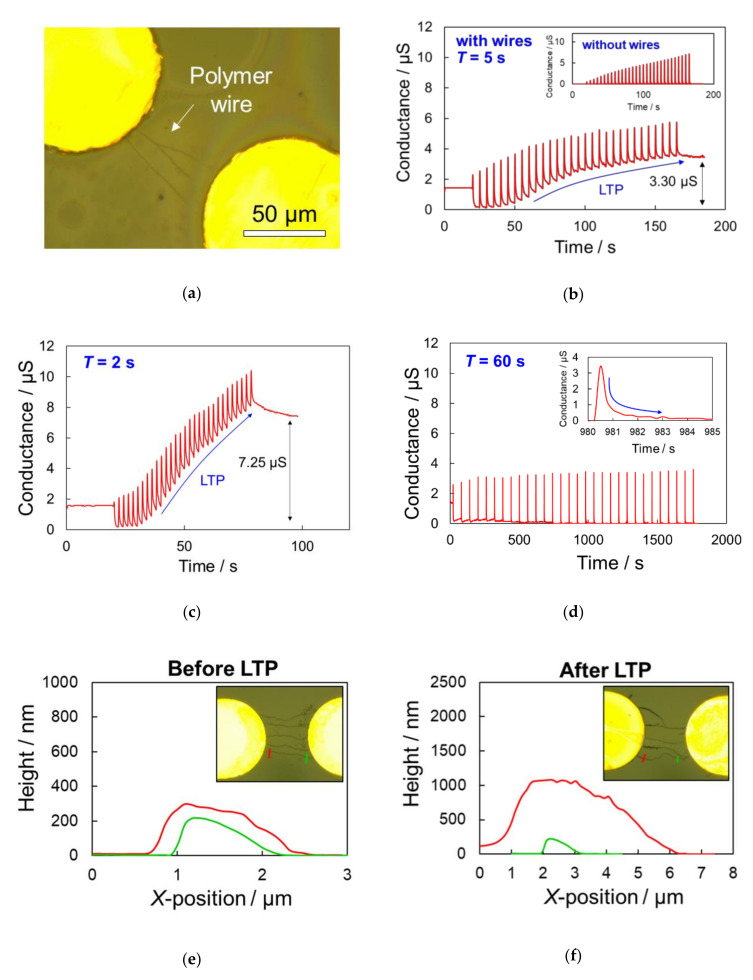
Expression of synaptic long-term potentiation (LTP) characteristics due to the application of periodic voltage pulsing (*V* = 2.5 V, *W* = 10 ms, 30 pulses) to an electrode gap linked by polymer wires. (**a**) Optical microscope image of the Au electrode gap and several bridging polymer wires before pulsing. (**b**) Change in conductance between the electrodes during continuous pulsing at *T* = 5 s, where the inset illustrates a control experiment of an electrode gap without any bridging polymer wires. (**c**,**d**) Change in conductance between the electrodes during continuous pulsing at *T* = 2 and 60 s, respectively. (**e**,**f**) Optical microscope images and atomic force microscopy (AFM) height profiles of the polymer wire before and after the expression of LTP, respectively.

**Figure 4 polymers-13-00312-f004:**
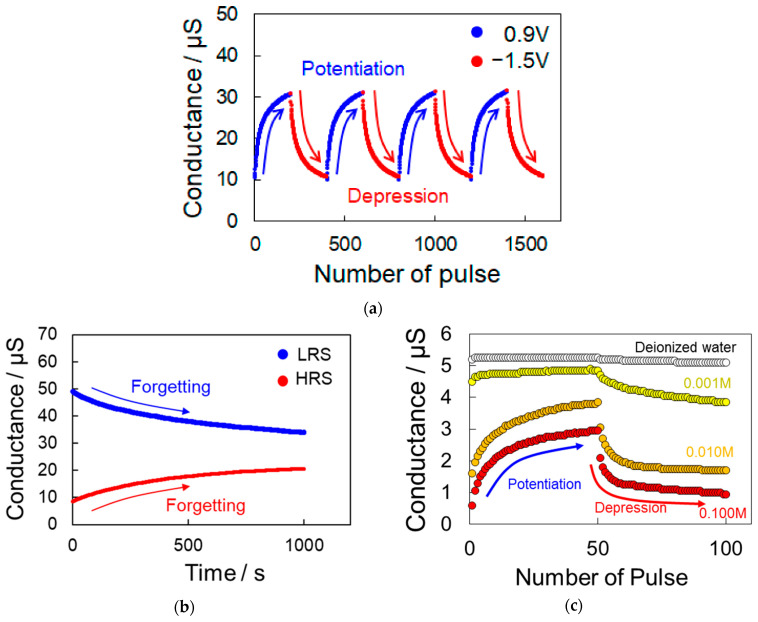
Expression of synaptic short-term plasticity (STP) characteristics due to the application of small periodic voltage pulses (*V* = 0.9 or −1.5 V, *W* = 10 ms, *T* = 0.5 s) to the electrode gap linked by polymer wires with asymmetric diameters. (**a**) Synaptic conductance change behaviors between the electrodes due to switching of the applied pulse polarity every 200 pulses. (**b**) State-retention curves for the polymer wire synapse with an open-circuit after being set to high-resistance state (HRS) or low-resistance state (LRS). (**c**) One cycle of synaptic conductance change behaviors due to switching of the applied pulse polarity every 50 pulses with increasing PSS dopant concentration from 0 to 0.1 M.

## Data Availability

The data presented in this study are available on request from the corresponding author.

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
