# Peer review of "Long- and Short-Term Conductance Control of Artificial Polymer Wire Synapses"

_polymers, 2021, doi:10.3390/polym13020312_

Round 1

Reviewer 1 Report

The authors report about the fabrication of PEDOT:PSS wires as wire synapses for long term potentiation and short term plasticity.

Their research is interesting and build upon previous results. While both STP and LTP effect clearly appear, I would like that the authors better explain the effect of the wire shape and PSS doping. In particular the effect of doping in the measurement environment, considering the amount of monomers and their concentration in the solution.

Also the ionic conductivity and different solubility of PSS/PSS- might have an effect on the measured conductivity, representing a more direct explanation than diffusion in the wire

Author Response

Thank you very much for providing important comments.

We are thankful for the time and energy you expended.

We attached a file with our responses to your comments.

Please see it.

Reviewer 2 Report

In this work, Hagiwara et al. developed bridging conductive wire based on PEDOT:PSS to control the conductance between to electrodes that has potential application in artificial intelligence. The results are interesting and the manuscript is well-written. It can be accepted in present form.  

Author Response

Thank you very much for providing important comments.

We are thankful for the time and energy you expended.

Reviewer 3 Report

The authors present nicely a manuscript regarding the possibility of use PEDOT:PSS as long/short-term conductance control material for wire synapses. The manuscript is well done, the introduction and the conclusions are consistent. The experimental part is clear and the result section is totally understandable, even to people with a different background.

I think this manuscript is acceptable as it is.

I have only some small inputs that could be corrected in the proof and that will complete the good work you have done:

-across the whole manuscript you use the word "crosslink"; actually I think could be substitute with "link" as the crosslink is intended as multi-branched system, in this case is more a bridge in between two objects.

-in line 58 you use this symbol "~" that means: in the same order of magnitude, if there is acceptable in line 148  "~0 μS" means that the value could be -5μS or +9μS and I think is not the case. Please use this symbol ≈ to refer to almost equality.

-line 226: the double pointed arrow in chemistry refer to a resonance, like in the benzene ring, for an equilibrium two arrows are used ⇋.

Author Response

(The authors gave the same response as above.)

Round 2

Reviewer 1 Report

I thank the authors for their reply.

Concerning my previous second comment:

If I understood correctly, the conductivity measurement where made in the same environment where the PEDOT:PSS wires were grown. Thus, there might still be monomers in the solution. This could cause an additional contribution to the conductivity, other than the wires themselves. Moreover, this could explain the thickening of the wires near the anodes.

I would appreciate if the authors could comment on this point, other than this I find the article suitable for publication in present form.

Author Response

Thank you very much for providing important comments.

We are thankful for the time and energy you expended.

In response to your comments, we've noticed that our explanations in the text were lacking  and some expressions were misleading , so we've corrected some of them.

If you find any inappropriate expressions in the text, please let us know.

Also, We attached a file with our responses to your comments.

Please see it.
